# Relationship between Anxiety and Problematic Smartphone Use in First-Year Junior High School Students: Moderated Mediation Effects of Physical Activity and School Adjustment

**DOI:** 10.3390/bs13110901

**Published:** 2023-11-01

**Authors:** Mei Cao, Haibo Yang, Duanduan Chen

**Affiliations:** 1Faculty of Psychology, Tianjin Normal University, No. 393 Binshuixi Road, Tianjin 300387, China; meicao@qfnu.edu.cn (M.C.); 2100340014@stu.tjnu.edu.cn (D.C.); 2School of Translation, Qufu Normal University, 80 Yantai North Road, Rizhao 276825, China

**Keywords:** apprehension, smartphone overuse, physical exercise, school adaptation

## Abstract

Background and aims: Despite previous research identifying anxiety as a risk factor for problematic smartphone use among students, the mediating and moderating mechanisms underlying the relationship between the two aforementioned variables are poorly understood. This study aims to explore the relationship between anxiety and problematic smartphone use among first-year junior high school students, together with the mediating effects of school adjustment and the moderating effects of physical activity on the mentioned relationship. Method: This study was conducted using a Web-based self-report questionnaire survey with data collected from 445 first-year junior high school students in Jinan City, Shandong Province. Mediation and moderation analyses were performed using the PROCESS macro in SPSS. Results: The results showed that anxiety predicted problematic smartphone use not only directly but also indirectly via school adjustment. School adjustment played a partial mediating role in the relationship between anxiety and problematic smartphone use. Physical activity also played a moderating role in the relationship between anxiety and school adjustment. Conclusion: school adjustment and physical activity may be important variables in the relationship between anxiety and problematic smartphone use.

## 1. Introduction

According to the 52nd Statistical Report on the Development Status of China’s Internet, the proportion of Internet users accessing the Internet via their smartphones reached 99.8% as of June 2023 in China. The data further show that underage Internet users in China have reached 183 million, and the Internet penetration rate is 94.9%, which is markedly higher than the figures for adult users. The proportion of underage Internet users owning devices with Internet access has reached 82.9%, with mobile smart terminals as the main Internet access devices [1].

The excessive use of smartphones can lead to many undesirable problems, such as addiction-like symptoms and feelings of dependency, which is termed problematic smartphone use [2]. Problematic smartphone use occurs when individuals suffer from impaired daily functioning as a result of smartphone use [3]. This condition can lead to numerous negative consequences, such as sleep deprivation [4] and family conflicts [5]. It can also lead to more serious problems, such as dangerous driving [6], loneliness [7] and depression [8]. Moreover, psychopathological symptoms such as depression [9], stress [10] and anxiety [11] can also be the antecedents of problematic smartphone use. In addition, a systematic review suggests that PSU often arises in tandem with emotional health issues such as depression, anxiety, anger and stress [12].

School adjustment is the psychological process by which an individual objectively recognizes the new environment and positively adapts his/her actions within the wider context of the school [13]. First-year junior high school students are in their critical period of physical and mental development, and school transition is an important phase they must undergo, but they are more likely to face problems with school adjustment [14]. During this stage, students are vulnerable to psychopathological symptoms such as decreased well-being, anger, depression and stress [15], and they are expected to experience a greater level of anxiety than usual [16].

Anxiety is related to problematic smartphone use among students [17]. Despite antecedent studies demonstrating the direct association between anxiety and problematic smartphone use, research on the mediating mechanism and moderating effect underlying this relationship remains inadequate. The present study aims to explore the mediator of school adjustment and the moderator of physical activity between anxiety and problematic smartphone use among Chinese students.

### 1.1. Anxiety and Problematic Smartphone Use

Anxiety, to a lesser extent, is related to problematic smartphone use. Consistent support has been found for anxiety severity and problematic smartphone use [18,19]. Anxiety is one of the variables predicting problematic smartphone use [20]. Specifically, excessive smartphone use is positively associated with anxiety [21].

The model of compensatory internet use indicates that anxiety can give rise to a motivation to go online to alleviate negative feelings [22], which may lead to problematic internet use. The theory has been supported by research on problematic smartphone use [23,24]. Thus, anxiety can significantly and positively predict problematic smartphone use [18]. The I-PACE model also indicates that anxiety is a predisposing variable, representing core characteristics of the person related to internet overuse (the P-components), and it is also one of the emotional responses induced by stress, which drives the individual to engage in internet overuse and problematic smartphone use [25].

### 1.2. School Adjustment as a Mediator

Individuals maladjusted to school may resort to problematic smartphone use as a way of coping [26]. Some researchers have suggested that school adjustment is a factor influencing problematic smartphone use and can significantly and negatively predict the level of such use [27]. Thus, a poorer level of school adjustment indicates a higher level of problematic smartphone use. However, the results of another study show that this peer relationship was only significantly predicted by smartphone overuse among high school students [28].

Meanwhile, school maladjustment has also been proposed as a consequence of problematic smartphone use—that is, individuals resort to smartphone use as an escape strategy from reality; the higher the level of smartphone among individuals, the greater the likelihood of developing school adjustment problems [29]. In addition, the dimensions of school adjustments, such as making peer relationships and keeping school rules, are also affected by smartphone overuse [30]. Thus, different results emerged for the subfactors of school adjustment and problematic smartphone use.

School maladjustment caused by anxiety may further contribute to problematic smartphone use [31]. For example, studies carried out among Korean 10-year-old children indicated that school adjustment results in problematic smartphone use [32]. Another study illustrated that school adjustment is positively correlated with problematic smartphone use [29]. Consequently, the present study posits that a high level of anxiety is related to a high level of school maladjustment, which further increases problematic smartphone use.

### 1.3. Physical Activity as a Moderator

Physical activity is defined as ‘any bodily movement produced by skeletal muscle that results in energy expenditure’ [33]. Physical activity alleviates the subsequent effects of anxiety faced by an individual and promotes school adaptation [34], which then reduces the level of problematic smartphone use among individuals [35,36]. Physical activity can negatively predict the level of problematic smartphone use [37,38]—that is, the higher the level of physical activity of an individual, the lower the level of problematic smartphone use [39,40].

Moreover, individuals with a high level of physical activity may not be susceptible to the negative effect of school maladjustment because they tend to regard themselves as energetic [41]. Thus, a high level of physical activity may buffer against school maladjustment by breaking its posited link with anxiety. By contrast, individuals with low levels of physical activity cannot mitigate the adverse effects of anxiety [42]. Adolescents’ participation in physical activity reduces school adjustment problems [43]. Participation in moderate-to-vigorous-intensity physical activity versus low-intensity physical activity is more conducive to reducing depressive symptoms associated with school adjustment [44].

### 1.4. Present Study

As indicated earlier, despite considerable research on anxiety and its destructive effects on adolescent mental health [45], its association with school adjustment among Chinese adolescents deserves attention because first-year junior high school students experience a change in the school environment from elementary school to secondary school and an increase in academic load; these students experiencing this transition can also develop school adjustment problems [14]. Moreover, the mediating role of school adjustment in the aforementioned association is less known, and the moderating effect of physical activity on the relationship between anxiety and school adjustment is yet to be explored. Aimed at gaining further insights into these mechanisms, this study proposes the following conceptual framework (in Figure 1) and following research hypotheses.

Individuals with higher level of anxiety demonstrate higher level of problematic smartphone use [46]. These individuals have social needs but are afraid of negative social experiences, such as rejection. Thus, they will spend time on mobile phones to keep in touch with others [47]. Therefore, anxiety may be positively related to problematic smartphone use. The first hypothesis is as follows:

**Hypothesis 1** **(H1).**
*Anxiety is positively related to problematic smartphone use.*


Individuals with high level of anxiety seem to face more problems with school adjustment [48], and they try to avoid face-to-face communication and alleviate negative emotions by overusing mobile phones, eventually producing problematic behaviors known as problematic smartphone use [49]. Therefore, the higher the level of anxiety is, the higher level of school maladjustment and the higher the level of problematic smartphone use. Based on the above analysis, the second hypothesis is as follows

**Hypothesis 2** **(H2).**
*A high level of anxiety is associated with a high level of school maladjustment, which leads to a high level of problematic smartphone use.*


The study has shown that individuals with a large amount physical activity may not be vulnerable to the negative consequences of school maladjustment because it can allow them to relax and better cope with changes in the school environment, resulting in improved school adjustment [50]. Based on the above analysis, the third hypothesis is as follows:

**Hypothesis 3** **(H3).**
*The indirect effect of anxiety on problematic smartphone use via school adjustment is moderated by physical activity. Specifically, the effect of anxiety on school adjustment problems is stronger for adolescents with a low level of physical activity than for those with a high level of physical activity.*


## 2. Method

### 2.1. Participants

A cross-sectional research design with four self-reported standardized questionnaires was used in this study, and convenience sampling was adopted in the selection of 499 first-year students from a town middle school in Zhangqiu District, Jinan City, Shandong Province. A questionnaire survey platform (https://www.wjx.cn/vm/tUttybZ.aspx#) (accessed on 2 February 2022) was used to collect responses about a week before the end of the winter holidays, from 6 February 2022 to 9 February 2022. The questionnaires were anonymous and confidential and were guided by guidance. With further screening, data with duplicate IPs were excluded, data with more than 95% of participants selecting the same answer were removed, data with filling times outside three standard deviations were removed and those with missing values for age were removed. Ultimately, 445 questionnaires were validated for the analysis. Thus, 445 participants (205 males) with a mean age of 13.86 years (SD = 0.79) were included in the analyses.

### 2.2. Measures

The Physical Activity Rating Scale, Smartphone Addiction Scale, Depression Anxiety Stress Short Version Scale and school Adjustment Scale were used in this study. It takes about 10 min to fill in the four scales.

#### 2.2.1. Physical Activity Rating Scale

Physical activity was measured using the Physical Activity Rating Scale. Developed by Liang, this scale examines the amount of physical activity of individuals. The 3-item self-reported scale consists of intensity, duration and frequency [51]. Each item is evaluated from 1 to 5, and the total score of physical activity is computed using the equation intensity × duration × frequency, which ranges from 0 to 100. Within this scale, the amount of physical activity individuals engage in is determined by the total sores. Specifically, a total score equal to 19 or below corresponds to a small amount of physical activity; 20–42 is defined as moderate physical activity, and 43 or above is considered a large amount of physical activity. A high score represents a high level of physical activity. In this study, the McDonald’s ω of the scale was 0.767.

#### 2.2.2. Smartphone Addiction Scale

Problematic smartphone use was measured by the Smartphone Addiction Scale. Kwon et al., developed this self-report scale to measure the level of smartphone addiction [52]. This 10-item self-report scale consists of the following components: health, social impairment, withdrawal and tolerance. Each item is evaluated from 1 (“not at all”) to 6 (“completely”). The scale operates on a scoring system where a higher score results in a higher index of smartphone addiction. The threshold for cell phone addiction varies for males (≥31) and females (≥33) [52]. In the current study, the version used was the Chinese version [53], and the McDonald’s ω was 0.930.

#### 2.2.3. Depression Anxiety Stress Short Version Scale

Anxiety was measured by the subscale of the Depression Anxiety Stress Short Version Scale, which includes 21 items that measure symptoms of depression, anxiety and stress. Lovibond et al., developed a self-report scale to measure the mental health of individuals [54]. Anxiety in first-year junior high school students was assessed using the anxiety subscale of the Depression Anxiety Stress Scale. The anxiety subscale consists of 7 items, and the 7 items are 2, 4, 7, 9, 15, 19 and 20. Each item is evaluated from 1 (“does not meet”) to 6 (“always meets”). In the current study, the Chinese version was used [55]. The McDonald’s ω of the anxiety subscale was 0.881.

#### 2.2.4. School Adjustment Scale

School adjustment was measured by the School Adjustment Scale. Cui developed a self-report scale to measure the level of school adjustment of middle-grade students. The school adjustment scale consists of 27 items and 5 dimensions: school attitude (7 items), peer relationship (6 items), teacher–student relationship (5 items), academic adjustment (7 items), and routine adjustment (4 items) [56]. Each item is rated on a five-point Likert scale ranging from 1 (“not at all”) to 5 (“completely”). On the positive scale, there are 3, 5, 14, 16, and 23 items, whereas on the negative scale, there are 22 items. A high score indicates a high level of school adjustment. In the current study, the McDonald’s ω was 0.943.

### 2.3. Data Analysis

Preliminary analyses (including normality of distribution, homogeneity of variances, missing cases, outliers and descriptive and correlational) of study variables (anxiety, school adjustment, physical activity and problematic smartphone use) were conducted using SPSS 24.0. Mediation and moderation analyses were performed in SPSS macro PROCESS. First, the mediation model was examined to identify the mechanism of school adjustment by which anxiety affects problematic smartphone use. The mediating effect was determined using 5000 bootstrap samples with a 95% confidence interval (CI). Second, the moderated mediation effects were evaluated. If the coefficient for the interaction between anxiety and physical activity was significant, the moderated mediation effect could be identified. Gender (1 = male, 2 = female), age, and class (1 = Class 1, 2 = Class 2, 3 = Class 3, 4 = Class 4, 5 = Class 5, 6 = Class 6, 7 = Class 7) were controlled as covariates in the regression models.

## 3. Result

### 3.1. Preliminary Analyses

The scores (anxiety, school adjustment, physical activity and problematic smartphone use) conformed to a normal distribution, variance chi-squared, and missing values and outliers (outside the three standard deviations) were removed. Table 1 presents the Pearson correlation coefficients, means, and SDs of the study variables. The results indicate that anxiety is positively associated with problematic smartphone use (*r* = 0.392, *p* < 0.01) but negatively correlated with school adjustment (*r* = −0.462, *p* < 0.01) and physical activity (*r* = −0.109, *p* < 0.05). School adjustment is negatively correlated with problematic smartphone use (*r* = −0.596, *p* < 0.01) but positively correlated with physical activity (*r* = 0.244, *p* < 0.01). Physical activity is negatively correlated with problematic smartphone use (*r* = −0.169, *p* < 0.05).

### 3.2. Mediation Analyses

Table 2 and Table 3 show that anxiety is directly and positively predictive of problematic smartphone use (*b* = 0.475, *p* < 0.001) and is negatively associated with school adjustment (*b* = −0.381, *p* < 0.001), which is negatively related to problematic smartphone use (*b* = −1.883, *p* < 0.001). The bootstrapped 95% CI confirms the significant indirect effects of school adjustment on the relationship between anxiety and problematic smartphone use (*b* = 0.718, 95% CI [0.510, 0.967]). These results indicate that school adjustment partially mediates the relationship between anxiety and problematic smartphone use.

### 3.3. Moderated Mediation Analyses

Table 4 and Table 5 present the results of the moderated mediation analysis when physical activity is entered as a moderator in the association between anxiety and school adjustment. The results suggest that the interaction between anxiety and physical activity significantly affects school adjustment (*b* = 0.006, *p* < 0.001). Thus, physical activity moderates the mediation of school adjustment in the association between anxiety and problematic smartphone use.

The effect of anxiety on school adjustment is examined using a simple main-effect analysis at one SD above and below the mean of physical activity. The interpretation of this moderating effect, with the predicted school adjustment value as a function of anxiety and physical activity, is shown in Figure 2. Simple slope tests reveal that the effect of anxiety on school adjustment is stronger for adolescents with a low level of physical activity (b = −0.485, *p* < 0.001) than for those with a high level of physical activity (b = −0.215, *p* < 0.001).

## 4. Discussion

This study explored the relationship between anxiety and problematic smartphone use, the mediating role of school adjustment, and the moderating role of physical activity on the association between anxiety and school adjustment among first-year junior high school students in China. This study found that anxiety positively predicted problematic smartphone use. Simultaneously, school adjustment exerted a partial mediating effect on the relationship between anxiety and problematic smartphone use—that is, anxiety predicted problematic smartphone use not only directly but also indirectly via school adjustment; in addition, the process of anxiety predicting problematic smartphone use through school adjustment was moderated by physical activity—that is, physical activity could moderate the process of anxiety predicting school adjustment. These three main findings are discussed as follows.

### 4.1. Relationship between Anxiety and Problematic Smartphone Use

The study found that anxiety positively predicted problematic smartphone use, thereby supporting H1. This finding is consistent with previous research showing that anxiety is a significant factor influencing problematic smartphone use [46]. Jin et al., demonstrated that the higher the level of anxiety, the higher the level of individuals’ problematic smartphone use [57].

Anxiety positively predicts smartphone overuse [58]. The reason is that being in a high-anxiety state leads to individuals having more difficulty communicating in real-life situations, resulting in the neglect of real-life interactions and an increase in anonymous communication on the Internet [59]. Cell phones provide relatively safe environments where individuals do not need to communicate, socialize or present themselves face-to-face in real life [48], thus leading to higher levels of problematic smartphone use in individuals with high anxiety levels. Moreover, anxiety can also be the consequences of smartphone overuse as individuals spend so much time on their smartphones while ignoring other important events in life [60]. In light of the current situation, it is essential to reduce the level of anxiety among first-year junior high school students as it can significantly predict problematic smartphone use.

### 4.2. Mediating Effect of School Adjustment on the Relationship between Anxiety and Problematic Smartphone Use

It has been revealed that school adjustment partially mediated the relationship between anxiety and problematic smartphone use in first-year junior high school students. This finding supports the hypothesis that anxiety significantly predicts problematic smartphone use through the mediation of school adjustment. Prior studies have also identified a link between anxiety and school adjustment as well as the effect of anxiety on reducing school adjustment [61,62]. School adjustment negatively predicted smartphone use, which is consistent with a previous finding that higher levels of school adjustment are associated with lower levels of individual problematic smartphone use. In a state of imbalance, students can feel negative emotions such as anxiety, dissatisfaction, and tension in the process of school adjustment, creating school adjustment problems [28]. Individuals with school maladjustment resort to avoiding realistic communication to solve their problems. Moreover, they use smartphones to mitigate negative emotional experiences associated with school maladjustment, which is consistent with the enrichment of the model of compensatory internet use and the I-PACE model. The theory of addiction recovery has identified anxiety as an important influencing factor leading to problematic smartphone use [63].

Echoed by those findings, this study found that students with higher amounts of anxiety tend to develop a greater level of school maladjustment, further contributing to their problematic smartphone use. In addition, anxiety indirectly predicts smartphone overuse via school adaptation, which not only enriches the mechanisms underlying the relationship between anxiety and smartphone overuse but also facilitates future targeted intervention research on individuals who excessively use smartphones.

### 4.3. Moderating Effect of Physical Activity on the Relationship between Anxiety and School Adjustment

Physical activity was found to have a moderating role in the relationship between anxiety and school adjustment. Further, anxiety was moderated by physical activity via the mediating process of school adjustment, predicting problematic smartphone use, thus corroborating H3. Specifically, individuals exhibiting higher levels of anxiety had lower levels of school adjustment compared with those exhibiting lower levels of anxiety for a given level of physical activity. Consistent with previous research findings, anxiety can affect the physical and mental health of an individual [64,65]. Stressful life events during adolescence are associated with various negative outcomes, such as decreased well-being, impaired mental health, anxiety and depression [15]

Other studies have further indicated that anxiety plays an important role in peer and teacher–student relationships among adolescents [66]. Thus, anxiety further affects the level of school adjustment of students [67]. Furthermore, this study supports that physical activity can alleviate the detrimental effect of anxiety on school adjustment. This result is consistent with previous studies that show that physical activity can be viewed as a protective mechanism [68,69]. Physical activity can also promote the release of dopamine and reduce the subsequent effects of anxiety [70]. In addition, physical activity can be a way for individuals to engage in recreation, allowing them to relax and better cope with changes in the school environment and increased academic stress, resulting in improved school adjustment [71]. In addition, the participation of adolescents in physical activity can reduce their school adjustment problems [42]. Compared with low-intensity physical activity, moderate-to-vigorous physical activity is more beneficial in reducing depressive symptoms associated with school adjustment [43]. Physical activity can also reduce the negative emotions caused by anxiety; thus, it moderates the relationship between anxiety and school adjustment [72]. In summary, the present findings support the effect of physical activity on the relationship between anxiety and school adjustment and expand its application to Chinese adolescent groups.

Overall, this study is an initial attempt to simultaneously integrate the mediating effect of school adjustment and the moderating effect of physical activity to interpret the association between anxiety and problematic smartphone use, particularly in the Chinese context. In addition, the result deepens the current understanding of the mechanisms through which anxiety influences problematic smartphone use.

The current study has several limitations. First, all subjects were first-year students from the same school and thus exhibited high homogeneity; the level of internal validity of the study was high, limiting the external validity of the study. The results of this study need to be replicated and extended in future research by adopting a method in which the diversity of the sample is increased to a higher level of external validity. Second, with regard to the research methodology, this study used a cross-sectional study that could not reveal a causal relationship between anxiety and problematic smartphone use. This causal relationship between anxiety and problematic smartphone use could be further explored in future research by manipulating anxiety and conducting a long-term follow-up to explore the dynamic relationship between anxiety and problematic smartphone use, school adjustment and physical activity. Finally, this study explored the mediating variables affecting the association between anxiety and problematic smartphone use and the moderating variables influencing the link between anxiety and school adjustment. Other mediating and moderating variables could be explored, and relevant theoretical models could be enriched in future research.

## 5. Conclusions

The following conclusions were drawn under the conditions of this study: School adjustment and physical activity may be important variables in explaining the relationship between anxiety and problematic smartphone use. The findings underline the significance of engaging in physical activity to reduce the level of school maladjustment and the problematic use of smartphones in clinical areas.

## Figures and Tables

**Figure 1 behavsci-13-00901-f001:**
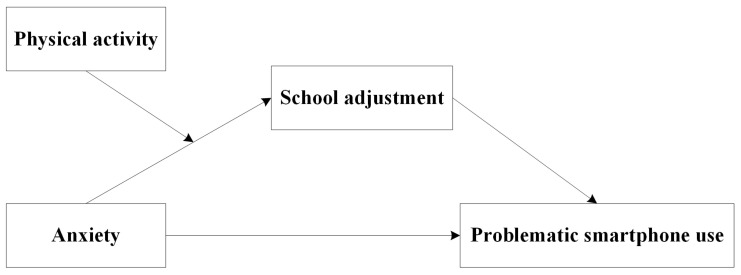
Conceptual framework.

**Figure 2 behavsci-13-00901-f002:**
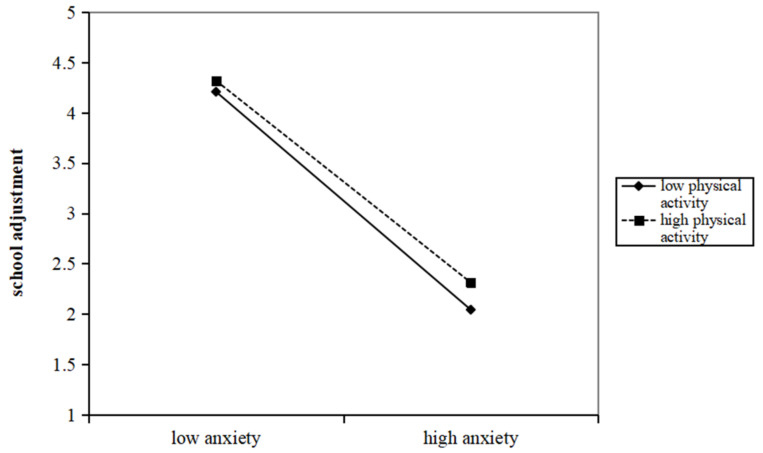
Physical activity as a moderator in the association between anxiety and school adjustment.

**Table 1 behavsci-13-00901-t001:** Descriptive statistics and correlations among variables.

	M	SD	1	2	3	4
1 anxiety	10.389	3.461	1			
2 school adjustment	22.206	2.854	−0.462 **	1		
3 problematic smartphone use	18.133	10.229	0.392 **	−0.596 **	1	
4 physical activity	27.342	22.585	−0.109 *	0.244 **	−0.169 *	1

Note: * *p* < 0.05; ** *p* < 0.01 (2-tailed).

**Table 2 behavsci-13-00901-t002:** Testing the mediation effect (Model 1 school adjustment).

Predictor	*B*	SE	*t*	*p*	95% CI
gender	0.403	0.242	1.662	0.097	[−0.074, 0.879]
age	−0.1765	0.154	−1.149	0.251	[−0.478, 0.125]
class	0.089	0.063	1.409	0.159	[−0.035, 0.213]
anxiety	−0.381	0.035	−10.909	0.000	[−0.450, −0.313]
school adjustment					
R^2^	0.223				
F	31.482 ***				

Note: *** *p* < 0.001.

**Table 3 behavsci-13-00901-t003:** Testing the mediation effect (Model 2 problematic smartphone use).

Predictor	*B*	SE	*t*	*p*	95% CI
gender	−1.3350	0.779	−1.713	0.087	[−2.867, 0.197]
age	−0.646	0.493	−1.311	0.191	[−1.615, 0.323]
class	0.087	0.202	0.429	0.668	[−0.311, 0.485]
anxiety	0.475	0.126	3.759	0.000	[0.227, 0.723]
school adjustment	−1.883	0.153	−12.321	0.000	[−2.183, −1.582]
R^2^	0.379				
F	53.768 ***				

Note: *** *p* < 0.001.

**Table 4 behavsci-13-00901-t004:** Testing the moderated mediation effect (Model 1: school adjustment).

Predictor	*B*	SE	*t*	*p*	95% CI
gender	0.570	0.238	2.400	0.017	[0.103, 1.037]
age	−0.207	0.148	−1.398	0.163	[−0.498, 0.084]
class	0.087	0.061	1.429	0.154	[−0.033, 0.207]
anxiety	−0.514	0.054	−9.452	0.000	[−0.621, −0.407]
physical activity	−0.032	0.018	−1.801	0.072	[−0.066, 0.003]
anxiety × physical activity	0.006	0.002	3.533	0.000	[0.003, 0.009]
R^2^	0.288				
F	29.509 ***				

Note: *** *p* < 0.001.

**Table 5 behavsci-13-00901-t005:** Testing the moderated mediation effect (Model 2 problematic smartphone use).

Predictor	*B*	SE	*t*	*p*	95% CI
gender	−1.629	0.733	−2.224	0.027	[−3.071, −0.189]
age	−0.646	0.493	−1.311	0.1901	[−1.6151, 0.323]
class	0.087	0.202	0.4297	0.668	[−0.311, 0.485]
anxiety	0.475	0.126	3.759	0.000	[0.227, 0.723]
school adjustment	−1.883	0.153	−12.321	0.000	[−2.183, −1.582]
R^2^	0.379				
F	53.768 ***				

Note: *** *p* < 0.001.

## Data Availability

The original data are available on reasonable request from the corresponding author.

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
