# Peer review of "Relationship between Anxiety and Problematic Smartphone Use in First-Year Junior High School Students: Moderated Mediation Effects of Physical Activity and School Adjustment"

_behavsci, 2023, doi:10.3390/bs13110901_

Round 1

Reviewer 1 Report

Comments and Suggestions for Authors

- The abstract is good.

- Keyword: Keywords different from those that appear in the title of the study must be used. This will allow it to be better accessible to the scientific community or readers if it is published. You should include synonyms or concepts linked or associated with those of the title but not exactly the same.

- Introduction: the concept “school adjustment” should be briefly defined and should not be taken for granted. It is known to have been expressed in line 62, but it is better to define it in the introduction, and describe or theorize about it later. It is a key term in this study, and cannot appear for the first time in the objective.

- 62-78. It may be relevant to add studies that show correlations between poor school adaptation and good use of mobile phones. That is, this study should reflect that there are perhaps other alternative uses of the mobile phone not associated with anxiety or other unhealthy emotions. Another option: define more specifically the adjective “problematic” in the concept “problematic use of smartphones”.

- 80. Reference 26 is a bit forced, given that it concerns a population with schizophrenia.

- 80. The statement “Physical activity promotes physical and mental health” should not always be true, because it will depend on the physical activity (type, quantity, quality, intensity, etc.). A definition of physical activity must be given in this study.

- 108. Check if the second hypothesis is correctly stated.

- 140. The reliability, or lack of reliability, associated with this Crobach index must be justified. On the other hand, Better results have been found measuring reliability with McDonald's omega (instead of alfa's Cronbach).

McDonald's omega. Please see Hayes & Coutts (2020) and McDonald's (1999).

References:

Hayes, A.F., & Coutts, J.J. (2020). Use omega rather than Cronbach's alpha for estimating reliability. But…. Communication Methods and Measures, 14(1), 1–24.

McDonald, R.P. (1999). Test theory: A Unified Treatment. Mahwah, NJ: Lawrence Erlbaum. https://doi.org/10.4324/9781410601087

It is advisable to use this reliability measure.

- 146. The period must come after the quotes.

- 149. It must be specified what this data refers to; that is, what it is measured by.

- 157. It must be specified what this data refers to; that is, what it is measured by.

- 163. There is a red dot.

-167. It must be specified what pre-analysis has been done: normality, homogeneity of variances, missing cases, outliers, etc.

- The results and discussion of this study are good. Congratulations.

- As at the beginning, the discussion should focus more on the specific Chinese population, given that it can socioculturally affect the interpretation.

The scientific effort of the author or authors is appreciated. However, there would be many improvements to be made.

Reviewer 2 Report

Comments and Suggestions for Authors

This is an interesting manuscript addressing an important topic. It is generally well-written; the method is sound and the conclusions are supported by results.

I only have some suggestions that could improve the paper:

- there is little reference to the fact that adolescence is an at-risk period for other psychopathological symptoms, other than anxiety and addiction. I suggest adding at least some hints about this.

- I read no reference to the possible weight of the COVID-19 pandemic on the study variables, although this topic is present in the reference list.

- in the introduction and in the discussion sections, I suggest stating more clearly (if this is the position of the authors) that psychopathological symptoms and internet addiction are mutually influencing, so that one can be precursor of the other. The authors report this position indirectly, and I suggest clarifying it.

- I suggest elaborating at list a bit on the conclusions, maybe considering the clinical implications of these results.

Comments on the Quality of English Language

The english is fine, with some errors and typos.
